# DUAL-AUGMENTATION GRAPH CONTRASTIVE PRETRAINING FOR TRANSISTOR-LEVEL CIRCUITS

## ABSTRACT

Structural information is the essence of graph data, and pretraining graph models aims to train neural networks that efficiently extract topological features. In electronic design automation (EDA), recent approaches use self-supervised graph pretraining to learn circuit representations and then fine-tune these pretrained models for downstream applications such as circuit performance prediction. However, due to limited circuit structures and the predominant focus on gate-level modeling, most existing methods can only extract structural information related to specific tasks or circuit types, such as digital circuits. To address this issue, we propose DICE: Device-level Integrated Circuits Encoder, a graph neural network (GNN) pretrained at the transistor level. DICE can model any circuits regardless of the signal they process, and is task-agnostic through graph contrastive learning using diverse circuit topologies. Our key contribution is the introduction of a dual-augmentation technique that generates over $10,000\times$ more topologies than prior work, thereby substantially increasing the structural diversity of circuits. Experimental results on three downstream tasks demonstrate significant performance gains on graph-level predictions, underscoring the effectiveness of DICE for circuit representation learning. Codes are released at [DICE-ICLR2026].

## 1 INTRODUCTION

Pretraining general-purpose models has demonstrated significant value across diverse domains. This includes large language models (LLMs) in natural language processing (Brown et al., 2020; Devlin et al., 2019), vision models in computer vision (Lu et al., 2019; Radford et al., 2021), and graph models in molecular science (Ross et al., 2022; Zeng et al., 2022). These models are applied to various downstream tasks beyond their original pretraining objectives and consistently deliver remarkable performance. Such success is largely attributed to the task-independent nature of unsupervised and self-supervised learning, which does not require expensive labeled data.

Pretraining task-agnostic models for integrated circuits (ICs) has long been an important problem in electronic design automation (EDA). ICs can be effectively represented as graphs, where circuit components (e.g., logic gates, transistors, or other devices) and their electrical connections (e.g., wires or nets) are modeled as nodes and edges. This graph-based abstraction enables the application of existing graph model pretraining methods to ICs. Prior work includes supervised (Hakhamaneshi et al., 2022; 2023; Shi et al., 2023), semi-supervised (Shi et al., 2024; Zheng et al., 2025), and self-supervised (Wang et al., 2022b; 2024b; Fang et al., 2025a;d) pretraining across different signal types (e.g., analog, mixed-signal, and digital). Particularly, self-supervised pretraining is a promising approach in EDA, as diverse circuit topologies are scarce and graph augmentation helps mitigate this issue. However, prior self-supervised work has focused primarily on digital circuits, and it remains unclear how to pretrain task-agnostic graph models independent of specific circuit types.

> **Then, how can we pretrain a model for integrated circuits (ICs) that is task-agnostic and effective regardless of the signals (e.g., analog, digital) they process?**

Without signals or simulations, the only available information is the topological structure of each circuit. Therefore, in this work, we investigate how pretraining a graph neural network (GNN) can facilitate the extraction of structural circuit features independent of specific tasks or signals.

Figure 1: **Limitations of prior work in pretraining circuit models.** (a) Logic gate–level pretraining cannot model analog and mixed-signal circuits, but transistor-level pretraining can model all. (b) Large datasets in transistor-level pretraining come from numerous device parameter combinations, not the diversity in circuit structures. (c) Conventional graph augmentations disrupt useful circuit semantics and therefore cannot be used for positive sample generation in graph contrastive learning.

To this end, we focus on the following three key aspects:

**(1) Unsupervised Pretraining (2) Transistor-Level Analysis (3) Topological Diversity**

First, unsupervised (or self-supervised) pretraining is necessary. Supervised pretraining (Ren et al., 2020; Hakhamaneshi et al., 2022; 2023; Shen et al., 2025b) requires costly circuit simulations to generate substantial labeled data, which are tightly coupled to specific simulation signals and device parameters. Since our goal is to develop general-purpose pretrained models rather than those tailored to particular simulation cases, supervised approaches fall outside the scope of this work.

Second, transistor-level representation is crucial for analyzing circuit structures, especially in analog and mixed-signal (AMS) designs. AMS circuits cannot be represented solely with logic gates and must be considered at the transistor level. Prior work on digital circuit pretraining represents logic gates as graph nodes (Wang et al., 2022b; 2024b; Shi et al., 2023; 2024; Zheng et al., 2025), and therefore cannot capture information from AMS circuits (Figure 1(a)). In contrast, transistor-level representation can also express digital circuits, since logic gates are composed of transistors, thereby providing a finer-grained and more expressive structural representation than gate-level abstraction.

Third, diverse circuit topologies are essential for extracting structural features. Pretraining with limited topologies may overfit to specific graphs, even under unsupervised or self-supervised learning (Shen et al., 2025a). Prior transistor-level work prepares substantial data by varying device parameters but lacks structural diversity, as illustrated in Figure 1(b).

**To the best of our knowledge, no prior work satisfies all three requirements.** Among the possible solutions, graph contrastive learning (You et al., 2020; Xu et al., 2021; Zhu et al., 2021b;a; Ju et al., 2024) is particularly promising due to its self-supervised nature with graph structure augmentations. However, for transistor-level circuits, traditional graph augmentation techniques (e.g., randomly adding nodes or edges) are unsuitable for generating positive samples, which are critical for contrastive learning. As illustrated in Figure 1(c), simply adding an edge to ground completely disrupts the original circuit semantics, even if the resulting graph remains structurally similar. While certain specialized tools (Wolf et al., 2013; Brayton & Mishchenko, 2010) can be used to augment logic gate–level digital circuits (Wang et al., 2022b; Li et al., 2022b), a different approach is required for transistor-level circuit augmentation.

Considering these challenges, we propose a graph contrastive learning framework for pretraining **DICE: Device-level Integrated Circuits Encoder**. DICE is a pretrained graph neural network (GNN) that extracts topological information from transistor-level circuits and does not require device parameter values or signal information. The contributions of our work are as follows:

**Contributions.** (1) We propose the first self-supervised pretraining framework for transistor-level circuits that considers only topologies without circuit signals. (2) We introduce a dual-augmentation technique for graph contrastive learning that increases structural diversity by more than $10,000\times$. (3) We demonstrate that the structural features extracted by DICE yield significant performance gains on both analog and digital signal-related tasks.

## 2 Preliminaries

### 2.1 Graph Contrastive Learning

Graph contrastive learning is one of the major classes of unsupervised graph representation learning methods. These approaches learn node, edge, or graph embeddings by maximizing the similarity between positive (similar) samples and minimizing it between negative (dissimilar) samples. To learn robust representations applicable to various tasks, they rely on data augmentation to diversify training data. Augmentations for the graph include adding nodes or perturbing edges within the graph.

One of the most widely used objectives in contrastive learning is the NT-Xent loss (Chen et al., 2020). Given a dataset of $N$ samples, each sample is augmented twice to produce $2N$ samples, forming $N$ positive pairs. For each sample $i \in 1, 2, \ldots, 2N$, let $\mathbf{z}_i$ denote its embedding and $\mathbf{z}_{j(i)}$ denote the embedding of its positive counterpart. The NT-Xent loss is then defined as:

$$\mathcal{L}_{\text{NT-Xent}} = \frac{1}{2N} \sum_{i=1}^{2N} -\log \frac{\exp\big(\text{sim}(\mathbf{z}_i, \mathbf{z}_{j(i)})/\tau\big)}{\sum_{k=1}^{2N} \mathbf{1}_{[k \neq i]} \exp\big(\text{sim}(\mathbf{z}_i, \mathbf{z}_k)/\tau\big)} \tag{1}$$

$\text{sim}(\cdot, \cdot)$ denotes a similarity function (e.g., cosine similarity), and $\tau$ is a temperature coefficient, typically set to 0.1. Minimizing Eq. (1) encourages the embeddings of positive pairs to cluster in the feature space while pushing apart the embeddings of negative pairs.

Some approaches maximize the similarity between positive pairs without relying on negative samples. SimSiam (Chen & He, 2021) is one such method, and the objective is given by:

$$\mathcal{L}_{\text{SimSiam}} = -\frac{1}{2} \left( \frac{\mathbf{p}_1^\top \text{sg}(\mathbf{z}_2)}{\|\mathbf{p}_1\|_2 \cdot \|\text{sg}(\mathbf{z}_2)\|_2} + \frac{\mathbf{p}_2^\top \text{sg}(\mathbf{z}_1)}{\|\mathbf{p}_2\|_2 \cdot \|\text{sg}(\mathbf{z}_1)\|_2} \right) \tag{2}$$

Here, $\mathbf{p}_1$ and $\mathbf{p}_2$ are the outputs of the predictor network given the feature inputs $\mathbf{z}_1$ and $\mathbf{z}_2$, respectively. The operator $\text{sg}(\cdot)$ denotes the stop-gradient operation, and $\|\cdot\|_2$ represents the L2 norm. The pair $(\mathbf{z}_1, \mathbf{z}_2)$ always constitutes a positive pair, and Eq. (2) maximizes cosine similarity accordingly.

In graph-based settings, contrastive learning can be applied at three levels: node, edge, and graph. Depending on the target task, the inputs $\mathbf{z}$ in Eqs. (1)-(2) can represent node, edge, or graph embeddings. In this work, we pretrained DICE using graph embeddings.

### 2.2 Graph Pretraining for Integrated Circuits

Prior work on pretraining graph models for ICs can be categorized into two types: gate–based and transistor–based. These two approaches differ in how they convert circuits into a graph.

Gate-based methods construct graphs by representing logic gates (e.g., INV, NAND, NOR) as nodes. These graphs form directed acyclic graphs (DAGs) (Mishchenko et al., 2006) and are used for digital circuits. Methods such as Shi et al. (2023; 2024) pretrain models by comparing logic functionalities and then transfer them to downstream tasks such as logic synthesis and Boolean satisfiability (SAT) solving. Likewise, Wang et al. (2022b; 2024b) employ graph contrastive learning to pretrain models that cluster functionally equivalent circuits and apply them to tasks such as arithmetic block identification and netlist classification. To address data scarcity, these works leverage logic synthesis tools such as Yosys (Wolf et al., 2013) and ABC (Brayton & Mishchenko, 2010), which generate diverse circuit structures while preserving identical functionality. More recently, large language models (LLMs) have been used to augment digital circuits (Liu et al., 2023; Chang et al., 2024) and further integrated into multimodal circuit representation learning (Fang et al., 2025b).

Transistor-based methods construct graphs using more fundamental device components (e.g., transistors, resistors) rather than logic gates. Although they can represent all circuit structures, prior work has mainly focused on tasks for analog and mixed-signal (AMS) circuits, where gate-based approaches are insufficient. Supervised pretraining in Hakhamaneshi et al. (2022; 2023) trains GNNs to predict DC voltage outputs of analog circuits, which can be categorized as a node-level prediction task. The learned representations are then transferred to predict other simulation metrics, such as the gain and bandwidth of operational amplifier circuits. However, self-supervised pretraining remains underexplored, partly due to the lack of reliable data augmentation techniques at the transistor level. While prior work has proposed augmentation strategies for AMS circuits (Deeb et al.,

2023; 2024), these augmentations are task-specific and primarily focused on analog designs. Large language models (LLMs) offer an alternative, as they can generate AMS circuits (Lai et al., 2024; 2025; Vungarala et al., 2024; Bhandari et al., 2024), but their effectiveness as a general-purpose augmentation tool remains uncertain.

# 3 DUAL-AUGMENTATION GRAPH CONTRASTIVE PRETRAINING

Building a graph contrastive learning framework at the transistor level presents several challenges. We review more work related to the following challenges in Appendix A.

> **Challenges.** (1) Reliable transistor-level circuit-to-graph conversion. (2) Effective graph structure augmentation at the transistor level. (3) Well-defined contrastive learning objective for transistor-level circuits. (4) Several downstream tasks for model evaluation.

Addressing these challenges, we propose a graph contrastive learning framework for pretraining **DICE: Device-level Integrated Circuits Encoder.** We first introduce our graph construction method for transistor-level circuits (Section 3.1), and describe our dual graph augmentation technique (Section 3.2). Next, we detail our contrastive learning objective (Section 3.3) and explain how we integrate DICE to solve three graph-level prediction tasks (Section 3.4).

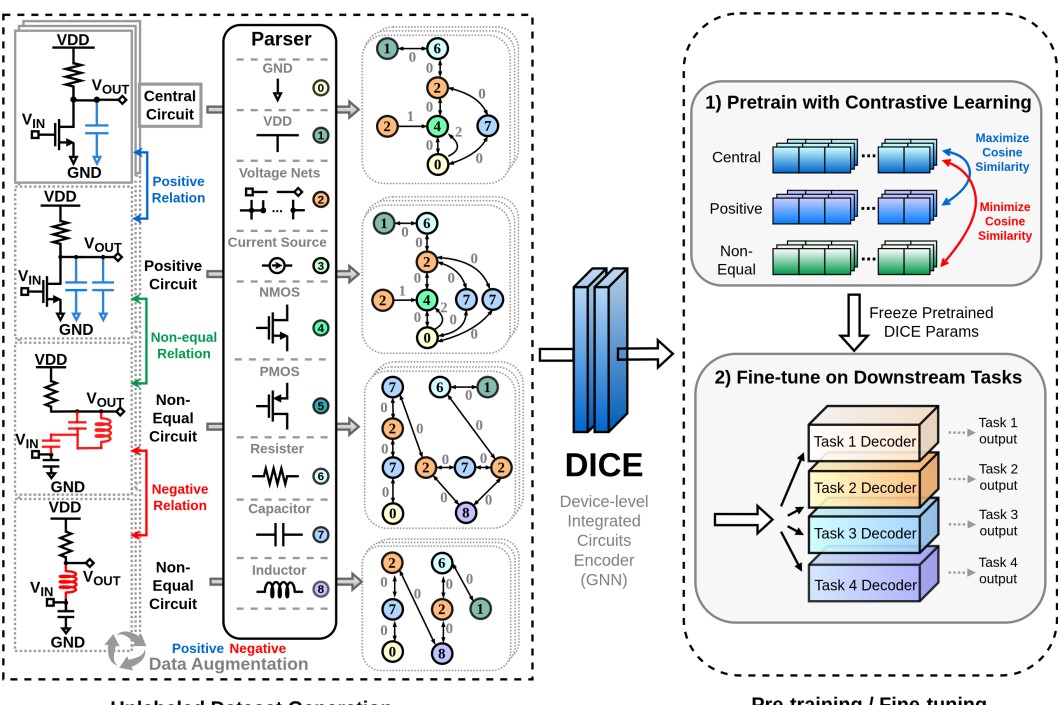

Figure 2: **Overview of our graph contrastive pretraining framework.** We maximize the cosine similarity between circuits in a positive relation and minimize it between circuits in a non-equal relation. A detailed explanation of these relations is provided in Section 3.3.

## 3.1 GRAPH CONSTRUCTION

As illustrated in the left box of Figure 2, we construct homogeneous graphs using one-hot encodings to represent device types and their connections. We define nine distinct node types: ground voltage net (0), power voltage net (1), other voltage nets (2), current source (3), NMOS transistor (4), PMOS transistor (5), resistor (6), capacitor (7), and inductor (8). In addition, we define five edge types to represent connections between devices: current flow path (0); link from a voltage net to the NMOS

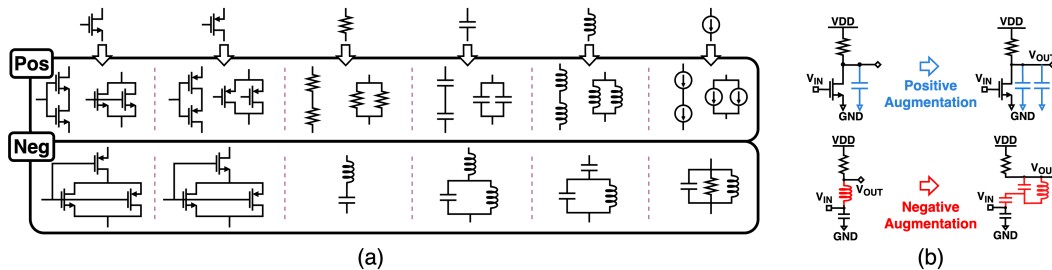

Figure 3: **Dual graph augmentation: (a) rules and (b) examples.** Positive augmentation adds an identical device in parallel or in series, preserving the overall circuit functionality. Negative augmentation replaces a subgraph, resulting in a circuit with different functionality.

gate (1) or bulk (2); and link from a voltage net to the PMOS gate (3) or bulk (4). Notably, edges are directed when connecting a voltage net to a transistor's gate or bulk node, since the influence of transistors on these nets is nearly negligible. With this design, our graph construction fully leverages edge features while assigning one node to each device.

## 3.2 DUAL GRAPH AUGMENTATION

To increase the diversity of circuit topologies, we introduce two data augmentation methods: positive and negative augmentation. This dual-augmentation is applied only to circuit structures and does not include any signal information or device parameters. For both augmentations, device subgraphs are modified within the full circuit graph, as illustrated in Figure 3(a). The theoretical details of our dual-augmentation are provided in Appendix B.

**Positive augmentation.** This method generates new circuit structures while preserving the graph-level functionality of the original circuits. In each iteration, a random device is selected, and an identical device is added either in parallel or in series. Although device parameters are not explicitly considered, this modification is functionally equivalent to altering the parameters of the selected device. As a result, the augmented graphs retain graph-level properties that closely resemble those of the original. Samples generated through positive augmentation can be further augmented to produce additional positive or negative samples. By maximizing the similarity between these positive samples, DICE learns to capture the underlying semantics shared across different circuit topologies.

**Negative Augmentation.** While positive augmentation increases structural diversity, it is limited in creating diverse graph-level functions. For example, if there are 50 distinct original circuits prior to augmentation, positive augmentation will generate new topologies whose graph-level properties remain constrained to these 50 functional types. As a result, the pretraining dataset would represent only 50 unique graph-level functionalities. To overcome this limitation, we introduce negative augmentation to enhance functional diversity. Specifically, we randomly select a device subgraph and replace it with an alternative subgraph, as illustrated in Figure 3(a). This structural modification induces significant changes in frequency response or DC voltage behavior once signal information is included, resulting in new topologies with distinct graph-level functionalities.

## 3.3 CONTRASTIVE LEARNING OBJECTIVE

After preparing the data with dual augmentation, we perform graph contrastive learning using graph-level embeddings of circuits, as illustrated in the upper-right part of Figure 2.

**Graph-level Embeddings.** To extract useful features from circuits, a GNN must first be defined. Accordingly, we design the architecture of DICE as a depth-2 Graph Isomorphism Network (GIN) that incorporates edge feature updates (Xu et al., 2018). The one-hot encoded embeddings from Section 3.1 are passed through DICE following Eqs.(4)–(6), resulting in graph-level embeddings.

**Positive, Negative, and Non-Equal Relations.** For contrastive comparison of embeddings, we define three types of relationships between circuits: positive, negative, and non-equal. First, two circuits are in a positive relation if both are generated through positive augmentation of the same original circuit. Second, two circuits are in a negative relation when one is positively augmented

and the other is negatively augmented from the same original circuit. Finally, all remaining pairs are categorized as non-equal, representing circuits that are augmented from different original circuits.

For example, consider two different original circuits, $x_1$ and $x_2$, prior to augmentation. Let $x_{1+}$, $x_{2+}$ denote circuits generated via positive augmentation from $x_1$ and $x_2$, respectively. Similarly, let $x_{1-}$ and $x_{2-}$ denote circuits generated via negative augmentation. In this case, positively related pairs include $(x_1, x_{1+})$ and $(x_2, x_{2+})$. Negatively related pairs include $(x_1, x_{1-})$, $(x_2, x_{2-})$, $(x_{1+}, x_{1-})$, and $(x_{2+}, x_{2-})$. Non-equal pairs include $(x_1, x_2)$, $(x_1, x_{2+})$, $(x_1, x_{2-})$, $(x_2, x_{1+})$, and $(x_2, x_{1-})$.

Our graph contrastive learning framework maximizes the cosine similarity between positive pairs and minimizes it only between non-equal pairs. We do not minimize the similarity between negative pairs, since these circuits differ only in localized structures and still share overall topologies that are more similar than those of non-equal pairs. Nevertheless, negative augmentation contributes to the set of non-equal pairs and provides additional diversity in graph-level functionalities.

**Contrastive Learning Objective.** We introduce a loss variant of Eq. (1), formulated in Eq. (3).

$$\mathcal{L}_{DICE}(\theta) = - \mathop{\mathbb{E}}_{x \sim p_{data}} \Big[ \frac{1}{N^+} \sum_{x' \in X^+(x)} \log \frac{\exp(f_\theta(x,x')/\tau)}{\sum\limits_{x^+ \in X^+(x)} \exp(f_\theta(x,x^+)/\tau_p) + \sum\limits_{x^{\neq} \in X^{\neq}(x)} \exp(f_\theta(x,x^{\neq})/\tau_n)} \Big] \quad (3)$$

The GNN feature extractor $f_\theta$, parameterized by $\theta$, computes the cosine similarity between the graph embeddings of two input graphs. The temperature coefficients are denoted by $\tau$, $\tau_p$, and $\tau_n$, and $p_{data}$ represents the probability distribution over all circuits in the pretraining dataset. Let $X^+(x)$ denote the discrete set of samples that are in a positive relation with $x$, and $X^{\neq}(x)$ denote the set of samples that are in a non-equal relation with $x$. We use $N^+$ to represent the number of samples in the set $X^+(x)$. Details on the practical implementation of Eq. (3) are provided in Appendix D.1.2.

### 3.4 EVALUATION ON DOWNSTREAM TASKS

After pretraining, we evaluate DICE on several downstream tasks to demonstrate that it is not restricted to specific simulations or signal types. We design three tasks: (1) circuit similarity prediction, (2) delay prediction on digital signals, and (3) op-amp performance prediction on analog signals. Each downstream task has multiple circuit topologies for structural diversity (Section 4.2).

For every downstream task, we fine-tune a complete model shown in Figure 4. The encoder (Figure 4(a)) comprises three GNN modules: two parallel networks that process the same graph input, followed by a series-connected network. One of the parallel GNNs is initialized and fixed with the pretrained DICE parameters, while the other parallel GNN and the series-connected GNN remain fully trainable. The encoder configuration remains fixed for each downstream task if the GNN depths are specified in advance. We denote the GNN depths of the two parallel networks and the series-connected network as $d_D$, $d_p$, and $d_s$, respectively.

The decoder network (Figure 4(b)) takes the output graph from the encoder as input and incorporates device parameters. To encode device parameters, each device type is represented by a 9-dimensional

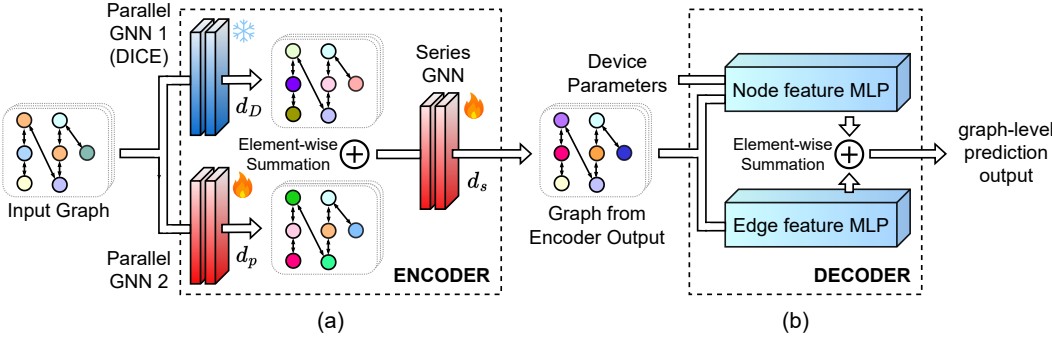

Figure 4: **Encoder (a)–Decoder (b) model architecture used for solving downstream tasks.** For Parallel GNN 2 and Series GNN, both node and edge features are updated according to Eqs.(4)–(6).

one-hot vector and scaled by its corresponding parameter values. Because parameter values can vary significantly in magnitude across device types, we apply a negative logarithmic transformation before scaling. The resulting encoded parameters are then concatenated with the node features from the encoder output and passed through the subsequent layers of the decoder.

**Takeaways.** (1) Homogeneous graph with edge direction. (2) Positive and negative graph augmentations. (3) Similarity minimization for non-equal pairs. (4) Fine-tune on three tasks.

## 4 EXPERIMENTS

We outline the experimental setup and results for both pretraining and three downstream tasks. Further experimental details are provided in Section D.

### 4.1 PRETRAINING

**Setup.** We compare three different contrastive learning approaches. The first approach uses $\mathcal{L}_{\text{SimSiam}}$ (Eq.(2)) with a pretraining dataset generated solely through positive augmentation. The second approach uses $\mathcal{L}_{\text{NT-Xent}}$ (Eq.(1)), also with a training dataset constructed using only positive augmentation. The third approach is our method described in Section 3, which uses $\mathcal{L}_{\text{DICE}}$ (Eq.(3)) and incorporates both positive and negative data augmentation. For $\mathcal{L}_{\text{NT-Xent}}$, we set $\tau = 0.05$. For $\mathcal{L}_{\text{DICE}}$, we set $(\tau, \tau_p, \tau_n) = (0.05, 0.2, 0.05)$. All GNNs in pretraining follow Eqs.(4)–(6).

**t-SNE Visualization.** Figure 5 presents t-SNE plots of the graph embedding vectors. We use the pretraining test dataset (Section D.1) constructed solely with positive augmentation, as negatively augmented samples cannot be colored the same as positive samples. Each of the 15 colors represents a distinct initial circuit, and all of its positively augmented views share the same color. Figure 5(c), (d), and (e) exhibit improved clustering compared to Figure 5(a) and (b), which indicate results without pretraining. This provides qualitative evidence of effective contrastive pretraining.

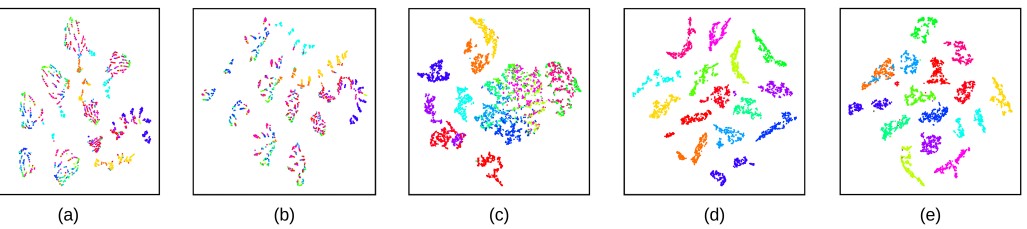

| (a) | (b) | (c) | (d) | (e) |

Figure 5: **t-SNE visualization of graph-level features.** Each plot represents the visualization of embeddings from (a) initial one-hot encodings, (b) untrained GNN, (c) GNN pretrained with $\mathcal{L}_{\text{SimSiam}}$, (d) GNN pretrained with $\mathcal{L}_{\text{NT-Xent}}$, and (e) GNN pretrained with $\mathcal{L}_{\text{DICE}}$.

**Cosine Similarity Values.** To further validate pretraining, we compute cosine similarities between circuit embeddings in the test dataset. For this, we used the dataset (Section D.1) constructed with both positive and negative augmentation. Table 1 presents the similarity values between positive, non-equal, and negative pairs. The results from DICE show that cosine similarities are high for positive pairs and are low for non-equal and negative pairs, validating the visualization in Figure 5.

Table 1: **Cosine similarities between circuit embeddings.** Mean / standard deviations are reported.

| Pairs | Initial | Untrained GNN | Pretrained GNN with $\mathcal{L}_{\text{SimSiam}}$ | with $\mathcal{L}_{\text{NT-Xent}}$ | DICE |
|---|---|---|---|---|---|
| Positive | 0.998±0.002 | 0.995±0.006 | 0.804±0.393 | 0.958±0.080 | **0.906**±0.161 |
| Non-equal | 0.993±0.005 | 0.982±0.016 | 0.124±0.708 | 0.481±0.350 | **0.349**±0.491 |
| Negative | 0.997±0.002 | 0.993±0.006 | 0.505±0.636 | 0.861±0.175 | **0.177**±0.362 |

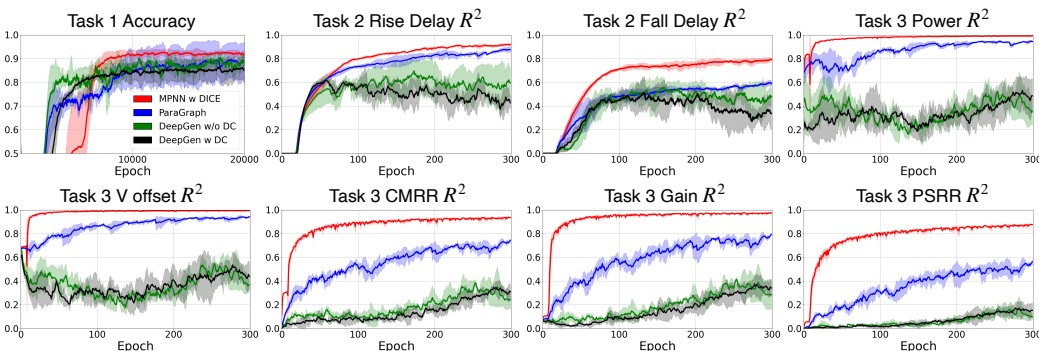

Figure 6: **Performance curves during downstream task training.** Each plot shows the $\mu \pm \sigma$ range across 3 different seeds. We compared our method (red), ParaGraph (blue), DeepGen$_u$ (green), and DeepGen$_p$ (black). Applying DICE leads to stable training and consistently outperforms all others.

## 4.2 DOWNSTREAM TASKS

To show how DICE extracts structural information well, we organize three downstream tasks. Each task handles multiple circuit topologies, requiring the GNNs to perform well on diverse structures.

**Downstream Task Settings.** For task 1, we train models to predict relative similarity between fifty different circuits, including both analog and digital circuits. For task 2, five delay line circuits are included, and the model predicts the signal delay values of a digital clock signal. For task 3, the models deal with five operational amplifier (op-amp) circuits, and the goal is to predict the performance values on analog signals. Further experimental details are provided in Appendix D.2.

**Baseline Setup.** Our baselines include prior work that (1) clearly defines transistor-level graph construction, (2) uses GNNs, and (3) predicts circuit simulation results. Specifically, we selected the un-pretrained GNN model (**ParaGraph**) from Ren et al. (2020), and both the unpretrained (**DeepGen$_u$**) and pretrained (**DeepGen$_p$**) GNN models from Hakhamaneshi et al. (2023) as baselines. DeepGen$_p$ is pretrained on a supervised DC voltage prediction task, with details provided in Appendix D.3. For each downstream task, the GNNs from the selected baselines are used as encoders (Figure 4(a)), while the decoder component (Figure 4(b)) remains identical across all models, including ours.

**Results.** We compare the baselines with our encoder model incorporating DICE (Figure 4(a)), using a GNN depth configuration of $(d_D, d_p, d_s) = (2, 0, 2)$. Figure 6 shows the validation performance curves of all encoder-decoder models during training. Final performance comparisons for each downstream task are summarized in Table 2. Each model was trained using three different random seeds, and we report the average performance across these models. Both Figure 6 and Table 2 demonstrate that our model outperforms all baselines across the downstream tasks.

Table 2: **Baseline performance on downstream tasks.** The mean values are reported across 3 different seeds, and the best results are highlighted in bold. DICE outperforms all baseline encoders.

| Baseline Encoders | Task 1 (%) | Task 2 ($R^2$) | | Task 3 ($R^2$) | | | | |
|---|---|---|---|---|---|---|---|---|
| | Accuracy | Rise Delay | Fall Delay | Power | $V_{\text{offset}}$ | CMRR | Gain | PSRR |
| ParaGraph | 91.42 | 0.9255 | 0.6373 | 0.9312 | 0.9585 | 0.7912 | 0.8488 | 0.6215 |
| DeepGen$_u$ | 85.02 | 0.8866 | 0.6715 | 0.8403 | 0.8478 | 0.5637 | 0.6813 | 0.3820 |
| DeepGen$_p$ | 84.65 | 0.9131 | 0.6875 | 0.4479 | 0.5190 | 0.3246 | 0.3454 | 0.0851 |
| DICE | **93.94** | **0.9468** | **0.8085** | **0.9916** | **0.9950** | **0.9374** | **0.9769** | **0.8809** |

## 4.3 ABLATION STUDIES

$\mathcal{L}_{\text{SimSiam}}$ **vs.** $\mathcal{L}_{\text{NT-Xent}}$ **vs.** $\mathcal{L}_{\text{DICE}}$. We applied other GNN architectures pretrained with $\mathcal{L}_{\text{SimSiam}}$ and $\mathcal{L}_{\text{NT-Xent}}$ in the encoder of Figure 4 and evaluated them on our downstream tasks. As shown in Table 3, both methods improve performance over the baselines in tasks 2 and 3.

**DICE vs. Diverse GNNs.** We also evaluated the effect of pretraining different GNN architectures, with results provided in Table 3. GCN (Kipf & Welling, 2016), GraphSAGE (Hamilton et al., 2017), GAT (Velickovic et al., 2017), and GIN (Xu et al., 2018) were pretrained using $\mathcal{L}_{\text{DICE}}$. Pretraining diverse GNNs following Appendix C also leads to performance gains.

Table 3: **Results of ablation studies (1) and (2). Bolded** values outperform all baselines in Table 2. Each pretrained GNN serves as part of the encoder in Figure 4, with $(d_D, d_p, d_s) = (2, 0, 2)$.

| Pretraining Loss | Pretrained GNN | Task 1 (%) | Task 2 ($R^2$) | | Task 3 ($R^2$) | | | | |
|---|---|---|---|---|---|---|---|---|---|
| | | Accuracy | Rise Delay | Fall Delay | Power | $V_{\text{offset}}$ | CMRR | Gain | PSRR |
| $\mathcal{L}_{\text{SimSiam}}$ | DICE | 85.88 | **0.9427** | 0.8044 | **0.9911** | **0.9951** | 0.9353 | 0.9762 | 0.8768 |
| $\mathcal{L}_{\text{NT-Xent}}$ | DICE | 76.73 | **0.9492** | 0.7706 | 0.9898 | 0.9947 | 0.9328 | 0.9757 | 0.8720 |
| $\mathcal{L}_{\text{DICE}}$ | GCN | 85.92 | 0.9113 | **0.7467** | **0.9909** | 0.9940 | 0.9314 | 0.9743 | 0.8662 |
| $\mathcal{L}_{\text{DICE}}$ | GraphSAGE | 79.35 | 0.8975 | 0.7786 | 0.9899 | 0.9947 | 0.9346 | 0.9756 | 0.8702 |
| $\mathcal{L}_{\text{DICE}}$ | GAT | 84.32 | **0.9314** | 0.7981 | 0.9862 | 0.9867 | 0.9130 | 0.9624 | 0.8004 |
| $\mathcal{L}_{\text{DICE}}$ | GIN | **93.88** | 0.9414 | 0.7931 | **0.9915** | 0.9947 | 0.9336 | 0.9757 | 0.8788 |
| $\mathcal{L}_{\text{DICE}}$ | DICE | 93.94 | 0.9468 | 0.8085 | 0.9916 | 0.9950 | 0.9374 | 0.9769 | 0.8809 |

**Training from Scratch vs. Applying DICE.** We compared two models: one trained from scratch and the other incorporating DICE. With both models following the architecture shown in Figure 4, the encoder remains fixed across all downstream tasks, and the decoder varies by task but is consistent across models within each task. $d_D$ is set to 0 for models without DICE, and $d_D$ is set to 2 for models with DICE. The number of trainable parameters is kept constant across models with identical $(d_p, d_s)$ values. Table 4 presents the experimental results, and all bolded and underlined values correspond to the model using DICE, except for a few cases when $(d_p, d_s) = (1, 1)$.

Table 4: **Results on ablation study (3).** For each $(d_p, d_s)$ configuration, we report results *without DICE* ($d_D{=}0$, ✗ in table) and *with DICE* ($d_D{=}2$, ✓ in table). **Bold** marks better per setting pair; underlined is best results overall.

| $(d_p, d_s)$ w/ DICE | Task 1 (%) | | Task 2 ($R^2$) | | | | Task 3 ($R^2$) | | | | | | | | | |
|---|---|---|---|---|---|---|---|---|---|---|---|---|---|---|---|---|
| | Accuracy | | Rise Delay | | Fall Delay | | Power | | $V_{\text{offset}}$ | | CMRR | | Gain | | PSRR | |
| | ✗ | ✓ | ✗ | ✓ | ✗ | ✓ | ✗ | ✓ | ✗ | ✓ | ✗ | ✓ | ✗ | ✓ | ✗ | ✓ |
| (0,0) | 76.91 | **92.13** | 0.7541 | **0.9352** | 0.5456 | **0.7885** | 0.9651 | **0.9769** | 0.8261 | **0.9820** | 0.4794 | **0.8865** | 0.3754 | **0.9402** | 0.2798 | **0.7535** |
| (0,1) | 74.84 | **93.41** | 0.7706 | **0.9310** | 0.5486 | **0.7805** | 0.9718 | **0.9904** | 0.8976 | **0.9876** | 0.6384 | **0.9167** | 0.6441 | **0.9659** | 0.5024 | **0.8132** |
| (1,0) | 77.90 | **93.43** | 0.7666 | **0.9294** | 0.5405 | **0.7820** | 0.9636 | **0.9849** | 0.8825 | **0.9868** | 0.6019 | **0.9147** | 0.6013 | **0.9634** | 0.4501 | **0.8061** |
| (1,1) | 84.68 | **94.54** | 0.9404 | **0.9417** | 0.8124 | 0.7873 | 0.9827 | **0.9847** | 0.9876 | 0.9872 | 0.9125 | **0.9147** | 0.9633 | 0.9627 | **0.8064** | 0.8061 |
| (0,2) | 86.28 | **93.98** | 0.9237 | **0.9402** | 0.7618 | **0.7851** | 0.9866 | **0.9876** | 0.9867 | **0.9874** | 0.9066 | **0.9165** | 0.9610 | **0.9641** | 0.7900 | **0.8055** |
| (2,0) | 88.57 | **94.01** | 0.9229 | **0.9231** | 0.7790 | **0.8134** | 0.9745 | **0.9918** | 0.9863 | **0.9950** | 0.9091 | **0.9371** | 0.9605 | **0.9769** | 0.7929 | **0.8776** |

## 4.4 Discussion

Our experiments yield four main insights: **(1) Pretrained GNNs outperform training from scratch**, confirming the value of transferable graph representations. For example, our model with DICE achieves over 40% higher accuracy than DeepGen$_u$ on Task 3. **(2) Contrastive pretraining proves more effective than supervised objectives** (Hakhamaneshi et al., 2023). In Task 1, our method outperforms DeepGen$_p$ by approximately 11%, indicating that task-agnostic unsupervised learning facilitates stronger generalization. **(3) The proposed objective in Eq.(3) produces more robust features than prior contrastive losses.** While Chen et al. (2020) demonstrated that similarity minimization is not essential, our results suggest that a carefully designed objective with augmentation can further enhance performance. **(4) Our pretraining framework generalizes well across diverse GNN architectures.** In Task 3, all pretrained variants outperform ParaGraph and DeepGen baselines, underscoring the robustness of our approach.

## 5 Conclusion

In this work, we introduced DICE, a pretrained GNN for transistor-level circuit representation. Our study demonstrates that device-level representations enable general and flexible modeling across circuits regardless of their signals. By incorporating contrastive pretraining and new circuit-specific augmentations, DICE achieves consistent gains on several tasks spanning diverse circuit topologies. We view this as a step toward general-purpose models that unify analog and digital circuit design.

## 6 ETHICS STATEMENT

All authors have read and adhered to the ICLR Code of Ethics. This work does not involve human subjects, personal data, or confidential industrial datasets. All circuits are public or synthetic data.

## 7 REPRODUCIBILITY STATEMENT

We provide full details of the model architecture, loss functions, and augmentation strategies in Section 3, with additional explanations in Appendix B, C, and D. Dataset construction, hyperparameters, and training settings are described in Section 3, 4, and Appendix D. Both qualitative (t-SNE visualizations) and quantitative (performance metrics) evidence are included. An anonymized implementation and scripts are available here to support reproducibility.

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

# Appendix

## A    RELATED WORKS

### A.1    GRAPH CONSTRUCTION FOR TRANSISTOR-LEVEL CIRCUITS

There are various approaches for converting transistor-level circuits into graphs. A key challenge lies in how to represent transistors as graph components. Ren et al. (2020) models both transistors and voltage nets as graph nodes and distinguishes connection types using multiple edge types. They adopt a heterogeneous graph and omit power and ground voltage nets as nodes. Studies in Hakhamaneshi et al. (2022; 2023) pretrain the GNN (DeepGen) proposed in Li et al. (2020) using a supervised node-level task: DC voltage prediction. Their graph construction does not differentiate edge types and instead separates device terminals into individual nodes, disjoint from their corresponding device nodes. Recent works in Gao et al. (2025a;b) follow a similar modeling approach, treating each device pin as a separate node. Wei et al. (2025) form a homogeneous graph, setting transistors as single nodes and the connections as undirected edges.

### A.2    GRAPH AUGMENTATIONS IN EDA

Different graph augmentation techniques have been explored for different circuit types. For digital circuits, prior work such as Wang et al. (2022b; 2024b) applies logic-preserving transformations to generate diverse circuit views, often using logic-synthesis tools like ABC (Brayton & Mishchenko, 2010) and Yosys (Wolf et al., 2013). They perform equivalence-preserving rewrites while ensuring that augmented circuits maintain identical logic functionality. For analog and mixed-signal (AMS) circuits, augmentation strategies instead focus on topology-aware edits, such as subgraph cropping or iterative modifications (Deeb et al., 2023; 2024), to improve robustness in recognizing and classifying schematic structures without altering low-level semantics. More recently, Gao et al. (2025a;b); Lai et al. (2024; 2025) have proposed synthesizing analog topologies, providing a form of augmentation that extends beyond deterministic rewrites.

### A.3    CONTRASTIVE LEARNING IN EDA

Contrastive learning has also gained attention in EDA for solving various tasks. At the gate and netlist level, Wang et al. (2022b; 2024b) adopt functionality-aware contrastive objectives by leveraging equivalence-preserving rewrites to form positive pairs, enabling the learned encoders to transfer to downstream tasks such as arithmetic block identification and netlist classification. Other approaches, such as Fang et al. (2025a;c), extend this idea by aligning multiple design stages, performing cross-stage contrastive learning to map RTL, netlist, and layout representations into a shared latent space. At the transistor level, Wang et al. (2024a); Shen et al. (2025a) apply contrastive learning to a single circuit design and use it for AMS circuit performance prediction.

### A.4    GRAPH-LEVEL EMBEDDING EVALUATION IN EDA

Predicting properties of the entire circuit can significantly reduce design time, as simulations are often time-consuming. To address this, prior work on digital circuits has focused on training models to predict power, performance, and area (PPA) (Li et al., 2022a; Fang et al., 2023; Du et al., 2024), timing violations (Guo et al., 2022; Gandham et al., 2024), routing congestion (Wang et al., 2022a; Zhao et al., 2024), and functional similarities between digital circuits (Alrahis et al., 2022; Bücher et al., 2022). For analog circuits, previous studies have trained models to predict performance metrics such as gain, bandwidth, and power consumption (Ren et al., 2020; Liu et al., 2021; Chen et al., 2021; Hakhamaneshi et al., 2023; Shahane et al., 2023; Wu & Savidis, 2023; Khamis & Agamy, 2024; Chae et al., 2024; Wang et al., 2024a; Poddar et al., 2024), and have also trained models to identify functional similarities between analog circuit blocks or transistor groups (Deeb et al., 2023; 2024; Kunal et al., 2023; Xu et al., 2024).

# B DETAILS OF DUAL GRAPH AUGMENTATION

## B.1 POSITIVE DATA AUGMENTATION DETAILS

We explain why positive data augmentation leads to the preservation of graph-level functionality. Consider two resistors with resistance values $r_1$ and $r_2$. When connected in series, these resistors can be equivalently replaced by a single resistor with a resistance value of $r_1 + r_2$. Similarly, for parallel connections, they can be replaced by a resistor with a resistance value of $\frac{r_1 r_2}{r_1 + r_2}$. This principle applies analogously to all other circuit components. Since device parameters do not affect structural circuit properties, we can consolidate series/parallel-connected devices into single components. Consequently, we can make positive samples consistently converge toward unified topological structures (in our case, 55 initial circuits) through these replacements.

Figure 7: Equivalent subgraphs generated through positive data augmentation. Different subgraph topologies indicate the same local functionality, and this leads to equal graph-level functions if all other structures are the same.

## B.2 NEGATIVE DATA AUGMENTATION DETAILS

We explain why negative data augmentation leads to the perturbation of graph-level functionality. For passive elements, we consider their impedance values at both DC (zero frequency) and the high-frequency limit, and modify them to have the inverted impedance characteristics at both extremes. For current sources, we replace them with passive elements, which are equivalent to converting the energy supplier to an energy dissipator. Finally, for transistors, we consider switching behaviors based on gate voltage inputs and connect the counterpart transistors in both parallel and series. Since either impedance characteristics, energy consumption, or switching behavior is inverted, the graph-level functionality is modified with our proposed negative data augmentation.

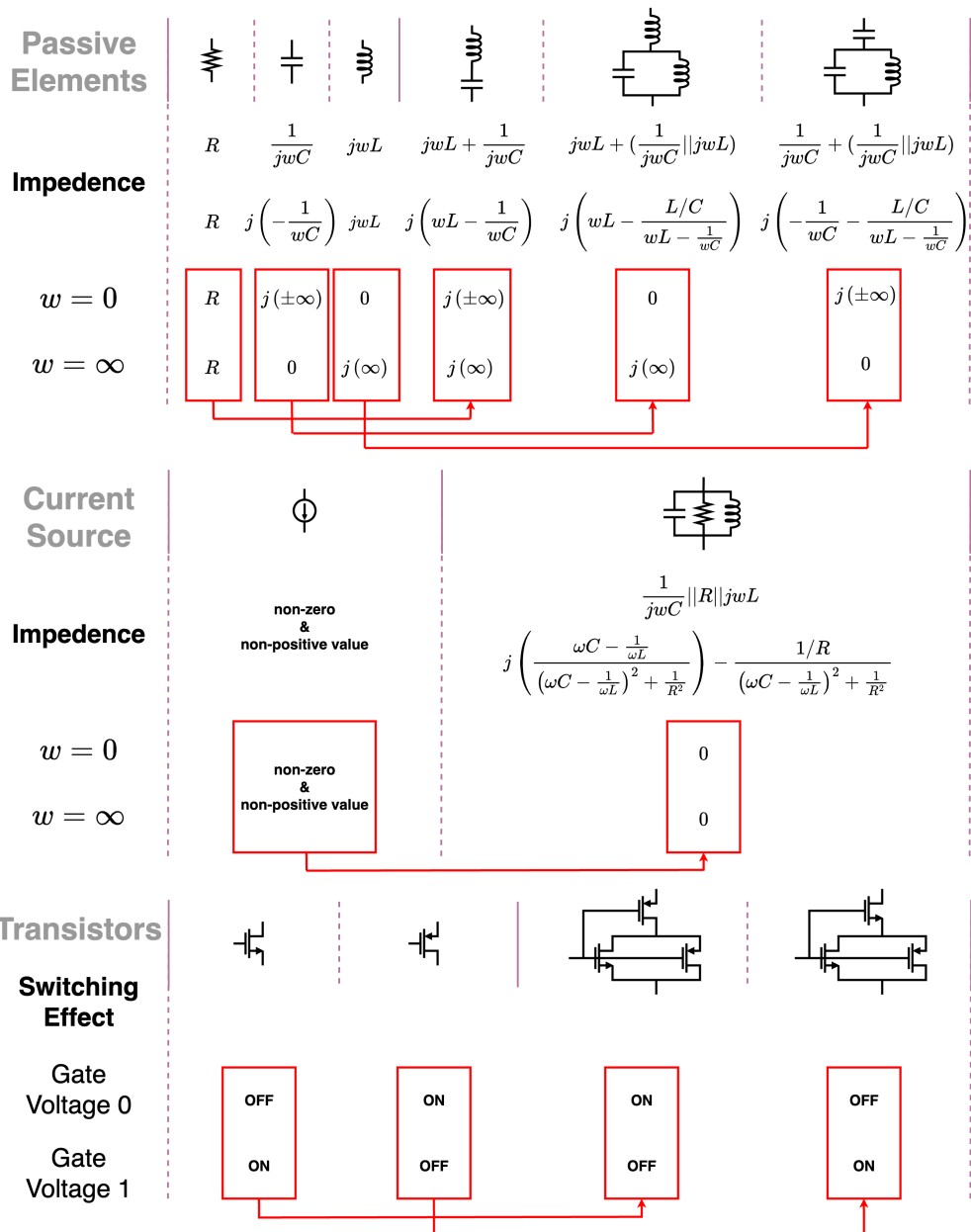

Figure 8: Conversion of characteristics through negative data augmentation.

# C    GNN UPDATE RULES

## C.1    DICE

The update rule of DICE follows the Graph Isomorphism Network (GIN) with edge feature updates.

$$\mathbf{h}_v^{(1)} = MLP_{\theta_0}(\mathbf{h}_v^{(0)}), \quad \mathbf{e}_{u \to v}^{(1)} = MLP_{\theta_1}(\mathbf{e}_{u \to v}^{(0)}) \tag{4}$$

$$\mathbf{m}_v^{(k)} = \sum_{u \in \mathcal{N}(v)} \mathbf{h}_v^{(k)} \cdot \mathbf{e}_{u \to v}^{(k)},$$

$$\mathbf{h}_v^{(k+1)} = MLP_{\theta_2^{(k)}}\left((1 + \phi_h^{(k)}) \cdot \mathbf{h}_v^{(k)} + \mathbf{m}_v^{(k)}\right), \tag{5}$$

$$\mathbf{e}_{u \to v}^{(k+1)} = MLP_{\theta_3^{(k)}}\left((1 + \phi_e^{(k)}) \cdot \mathbf{e}_{u \to v}^{(k)} + \mathbf{m}_u^{(k)} - \mathbf{m}_v^{(k)}\right)$$

$$\mathbf{g}_{(V,E)} = \sum_{v \in V} \mathbf{h}_v^{(l+1)} + \sum_{u \to v \in E} \mathbf{e}_{u \to v}^{(l+1)} \tag{6}$$

Eqs. (4)–(6) illustrate the node and edge feature update rules of DICE, where $\theta_0, \theta_1, \theta_2^{(k)}, \theta_3^{(k)}, \phi_h^{(k)}$, and $\phi_e^{(k)}$ are the training parameters. Batch normalization and dropout layers are included between linear layers. Eq. (4) matches the initial dimensions of node and edge features (9 and 5) to the hidden dimension value, and $\mathbf{h}_u^{(0)}, \mathbf{e}_{v \to u}^{(0)}$ are the initial one-hot encoding of node and edge types. Eq. (5) explains the message passing rule, where $k = 1, ..., l$ and $l(= 2)$ is the depth of the GNN. The graph-level feature $\mathbf{g}_{(V,E)}$ is calculated in Eq. (6), by summing all the updated node and edge features within each graph $G = (V, E)$.

## C.2    GCN

The update rule of the Graph Convolutional Network (GCN) used in this work follows a normalized aggregation mechanism:

$$\mathbf{m}_v^{(k)} = \sum_{u \in \mathcal{N}(v)} \frac{1}{\sqrt{d_v d_u}} \cdot \mathbf{h}_u^{(k)}$$

$$\mathbf{h}_v^{(k+1)} = \text{MLP}_{\theta^{(k)}}(\mathbf{m}_v^{(k)}) \tag{7}$$

Here, $d_v$ and $d_u$ denote the degrees of nodes $v$ and $u$, respectively. The GCN layer performs degree-normalized aggregation followed by a learnable transformation of the aggregated features. Although edge features are not explicitly involved in the message-passing process for GCN, Eq. (4) is still applied at the initial layer to project both node and edge features into a common hidden space. Eq. (6) is used to compute the graph-level feature by summing the final node and edge embeddings.

## C.3    GRAPHSAGE

GraphSAGE in our work performs aggregation by concatenating the central node's embedding with the mean of its neighbors' embeddings:

$$\mathbf{m}_v^{(k)} = \frac{1}{|\mathcal{N}(v)|} \sum_{u \in \mathcal{N}(v)} \mathbf{h}_u^{(k)}$$

$$\mathbf{h}_v^{(k+1)} = \text{MLP}_{\theta^{(k)}}\left([\mathbf{h}_v^{(k)} \| \mathbf{m}_v^{(k)}]\right) \tag{8}$$

Aggregation is performed through mean pooling, and node features are updated by concatenating the aggregated messages with the node's current embedding. Although edge features do not contribute to the message-passing step, Eq. (4) is used to initialize the edge feature embeddings. The final graph-level feature is computed via Eq. (6), which includes both node and edge representations.

## C.4 GAT

The Graph Attention Network (GAT) used in our work computes attention coefficients for each edge to weigh the contributions of neighboring nodes:

$$
\begin{aligned}
\alpha_{u \to v}^{(k)} &= \text{softmax}_{u \in \mathcal{N}(v)} \left( \mathbf{h}_u^{(k)} \cdot \mathbf{h}_v^{(k)} \right) \\
\mathbf{m}_v^{(k)} &= \sum_{u \in \mathcal{N}(v)} \alpha_{u \to v}^{(k)} \cdot \mathbf{h}_u^{(k)} \\
\mathbf{h}_v^{(k+1)} &= \mathbf{h}_v^{(k)} + \mathbf{m}_v^{(k)}
\end{aligned}
\tag{9}
$$

The attention mechanism allows the model to learn the relative importance of neighboring nodes based on their feature similarity. While edge features are not updated during message passing in GAT, they are initially transformed using Eq. (4). The final graph-level representation is computed by summing both the node and edge embeddings as described in Eq. (6).

## C.5 GIN

The update rule of the Graph Isomorphism Network (GIN) uses sum aggregation followed by a learnable MLP:

$$
\begin{aligned}
\mathbf{m}_v^{(k)} &= \sum_{u \in \mathcal{N}(v)} \mathbf{h}_u^{(k)} \\
\mathbf{h}_v^{(k+1)} &= \text{MLP}_{\theta^{(k)}} \left( (1 + \epsilon^{(k)}) \cdot \mathbf{h}_v^{(k)} + \mathbf{m}_v^{(k)} \right)
\end{aligned}
\tag{10}
$$

Here, $\epsilon^{(k)}$ is a learnable scalar, initialized as zero. GIN is known for its strong expressive power in distinguishing graph structures. Although the edge features are not used in message passing for GIN, they are projected to the hidden space using Eq. (4). Both node and edge embeddings are summed to form the graph-level representation, as defined in Eq. (6).

## D DETAILS OF THE EXPERIMENTS

### D.1 PRETRAINING

#### D.1.1 DATASET

Data augmentation is used exclusively for generating the pretraining datasets and is performed independently for the pretraining training and test sets. The pretraining dataset comprises 55 initial circuit topologies, covering both analog and digital designs. For the training set, 40 topologies are each augmented 4,000 times (2,000 positive and 2,000 negative), resulting in a total of 160,000 graphs. For testing with t-SNE visualization, the remaining 15 topologies are each augmented 500 times using only positive augmentation, yielding 7,500 graphs. For testing cosine similarity values, the same 15 topologies are each augmented 500 times (250 positive and 250 negative), also resulting in 7,500 graphs. We did not use negative augmentation for t-SNE visualization since samples with negative augmentation cannot be categorized and distracts the visualization results.

#### D.1.2 CONTRASTIVE LEARNING

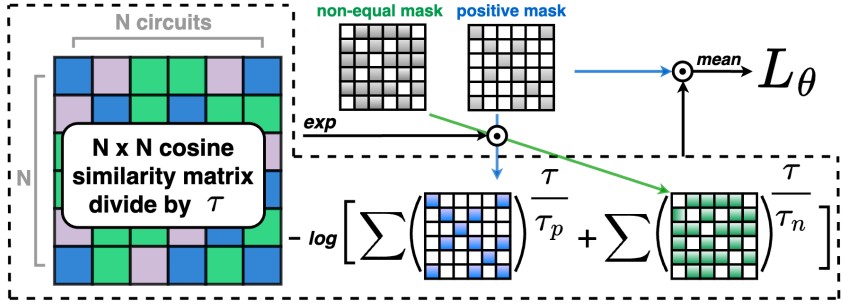

Figure 9: **Masking operation.** $\odot$ indicates the hadamard product (element-wise multiplication).

Masking is our key source of rapid pretraining, with the cost of GPU memory. Algorithm. 1 provides the pseudocode, and Fig. 9 visualizes the details of the proposed masking for contrastive learning. For each batch, we first matched the number of graphs for every circuit type. For example, consider the case when the number of graphs originated from the inverter circuit is 5. If this is the minimum number among all other circuit types in the batch, then 5 circuits are sampled for every circuit type and form a new batch. Then, the cosine similarity matrix is formed with graph-level features in the batch with masks indicating positive and non-equal. Element-wise products between the similarity matrix and the masks filter out the unnecessary parts in parallel. Multiple positive pairs are considered simultaneously with this operation, thereby significantly reducing the training speed at the cost of GPU memory.

---

**Algorithm 1** Graph-level Contrastive Learning

---

**Input:** augmented dataset $(\chi)$, epoch $(T)$, temperature coefficients $(\tau_p, \tau, \tau_n)$, learning rate $(\eta)$
**Output:** pretrained model parameter $(\theta_T)$
**Initialize:** model parameter $\theta_0$
1: **for** $t = 0, 1, 2, ..., T - 1$ **do**
2:   **for** batch $B$ in $\chi$ **do**
3:     $m \leftarrow$ min number covering all circuit types in $B$
4:     $B_m \leftarrow$ batch$\in \mathbb{R}^L$ having $m$ graphs each for every circuit types (maintaining circuit diversity)
5:     $g \leftarrow GNN_{\theta_t}(B_m)$: graph-level features $\in \mathbb{R}^{L \times h}$
6:     $S \leftarrow gg^T$: cosine similarity matrix $\in \mathbb{R}^{L \times L}$
7:     $M_+, M_{\neq} \leftarrow$ Mask matrices $\in \mathbb{R}^{L \times L}$ for positive and non-equal pairs
9:     $L_{\theta_t} \leftarrow \sum (M_+ * \left[ S/\tau - \log(\sum M_+ * e^{S/\tau_p} + \sum M_{\neq} * e^{S/\tau_n})\right])/ \sum M_+$
10:     $\theta_{t+1} \leftarrow \theta_t - \eta\nabla_{\theta_t} L_{\theta_t}$
11:   **end for**
12: **end for**

---

## D.2 Downstream Tasks

### D.2.1 Task 1: Circuit Similarity Prediction

**Setup.** The model predicts the relative similarities between three circuits, including both analog and digital topologies. The objective is to maximize prediction accuracy (%) based on true similarity, which is determined by the number of shared labels. A total of 6 labels include analog, digital, delay lines, amplifiers, logic gates, and oscillators.

Given a target circuit and two comparison circuits (a total of three), the model outputs a three-dimensional probability distribution: whether the first circuit is more similar, the second circuit is more similar, or the two circuits are equally similar to the target circuit. In each training step, one target circuit from the training dataset is sampled, and all permuted pairs across the $N$ number of circuits form a tensor of size $(N \times (N-1) = 2950, \ 3 \times graph\ feature\ dimension)$ for the training model. Based on the number of shared labels, the logits indicating similarity comparisons are computed. Using these logits and the model output, the cross-entropy loss is minimized.

**Dataset.** The training dataset consists of 50 circuits, including the 40 topologies from the pretraining dataset. The test dataset contains 5 additional circuits, resulting in a total of 55 circuits.

### D.2.2 Task 2: Delay Prediction

**Setup.** The model predicts simulation results for five delay line circuits, which operate on digital signals. Each simulation records two values: the time difference between the rising edges and the falling edges of the input and output clock signals. The objective is to maximize the coefficient of determination ($R^2$) across all simulations, which measures how well the model's predictions match the true simulation results.

**Dataset.** For simulation, we used NGSPICE with the BSIM4 45nm technology, which serves as the open-source circuit simulator and technology file, respectively. The delay line circuits used in Task 2 are not included in the pretraining dataset. The simulation result dataset contains 45,000 device parameter combinations for each circuit topology. We split the dataset into training, validation, and test sets with a ratio of 8:1:1. The output dimension of the decoder model is 2.

### D.2.3 Task 3: OPAMP Metric Prediction

**Setup.** The model predicts simulation results for five operational amplifier (op-amp) circuits, which are analog circuits. Each simulation records five metrics: power, DC voltage offset ($V_{offset}$), common-mode rejection ratio (CMRR), gain, and power supply rejection ratio (PSRR). The objective is to maximize the coefficient of determination ($R^2$) across all simulations, which evaluates how well the model predictions align with the true simulation results.

**Dataset.** NGSPICE with the BSIM4 45nm technology file is also used for Task 3. The op-amp circuit topologies used in Task 3 are included in the pretraining dataset, but the simulation results are newly generated for this task. The simulation dataset contains 60,000 device parameter combinations for each topology. The circuit and testbench files are sourced from Li et al. (2024). The dataset is split into training, validation, and test sets using an 8:1:1 ratio. The decoder network used in Task 3 is identical to that in Task 2, except that the output dimension is set to 5.

### D.3 PRETRAINING WITH DC VOLTAGE PREDICTION TASK

We pretrained DeepGen$_p$ using the DC voltage prediction task proposed in Hakhamaneshi et al. (2022; 2023). In the original work, the authors used two circuit topologies for pretraining: a resistor ladder and a simple operational amplifier (op-amp). They also trained separate GNNs for each topology.

However, to evaluate whether this supervised pretraining approach generalizes effectively, the pretraining process should include more than two circuit topologies simultaneously. In our setting, we pretrained a single GNN on seven different circuit topologies: one resistor ladder, one current mirror, three simple RLC circuits, and two op-amps. We used the same GNN architecture described in Hakhamaneshi et al. (2022; 2023) to ensure a fair comparison.

### D.4 HYPERPARAMETERS

| Training parameters | DICE Pretraining | SimSiam Pretraining | NT-Xent Pretraining | DeepGen Pretraining | Downstream Task 1 | Downstream Task 2 | Downstream Task 3 |
|---|---|---|---|---|---|---|---|
| learning rate | 0.0003 | 0.0003 | 0.0003 | 0.000005 | 0.00001 | 0.0001 | 0.0001 |
| batch size | 1024 | 1024 | 1024 | 1024 | 50 | 2048 | 1024 |
| epochs | 200 | 200 | 200 | 200 | 20000 | 300 | 300 |
| $(\tau_p, \tau, \tau_n)$ | (0.2, 0.05, 0.05) | (–,0.05,–) | – | – | – | – | – |
| optimizer | Adam | Adam | Adam | Adam | Adam | Adam | Adam |

| Hyperparameters | DICE | Encoder | Decoder | DeepGEN | ParaGraph |
|---|---|---|---|---|---|
| initial node feature dimension | 9 | – | – | 15 | 7 |
| initial edge feature dimension | 5 | – | – | 1 | 5 |
| hidden dimension | 256 | 512 | 256 | 512 | 512 |
| dropout probability | 0.2 | – | 0.3 | – | – |
| activation | GELU | GELU | GELU | GELU | GELU |
| GNN type | GIN | GIN | – | GAT | HGAT |
| GNN depth | 2 | depends | 0 | 3 | 3 |

### D.5 COMPUTE RESOURCES

All experiments were conducted on a server running Ubuntu 20.04.6 LTS with Linux kernel 5.15. The machine is equipped with an Intel(R) Core(TM) i7-9700 CPU operating at 3.00 GHz, 62 GB of RAM, and a single NVIDIA Quadro RTX 6000 GPU with 24 GB of VRAM.

The DICE pretraining stage was executed on a single RTX A6000 GPU and required approximately 2 hours of wall-clock time per trial. Each downstream task was run on a single GPU as well, with Task 1 taking around 1 hour, Task 2 approximately 3 hours, and Task 3 about 8 hours per trial.

### USE OF LARGE LANGUAGE MODELS (LLMS)

An LLM was used only for language polishing (e.g., grammar and phrasing).

