# OpenReview forum: "Dual-Augmentation Graph Contrastive Pretraining for Transistor-level Integrated Circuits"
_ICLR.cc/2026/Conference — ICLR 2026 Conference Withdrawn Submission_

### Official Review · Reviewer_NggG · 2025-10-26

**Soundness:** 2
**Presentation:** 2
**Contribution:** 2
**Rating:** 2
**Confidence:** 4

**Summary:**

This paper introduces DICE (Device-level Integrated Circuits Encoder), a graph neural network (GNN) pre-trained using self-supervised contrastive learning on transistor-level circuit representations. The authors argue that existing pre-training methods in Electronic Design Automation (EDA) suffer from limitations: (1) they often operate at the gate level, making them unsuitable for analog and mixed-signal (AMS) circuits, (2) existing transistor-level methods often rely on supervised pre-training tied to specific simulations, limiting generality, and (3) self-supervised methods lack sufficient structural diversity and appropriate augmentation techniques for transistor-level graphs, as simple augmentations break circuit semantics.

DICE aims to overcome these issues by Transistor-Level Graph Representation, Dual-Augmentation Technique and Contrastive Learning Objective.

**Strengths:**

Novel Dual-Augmentation: This is the core technical contribution and a clever solution to a difficult problem. Standard graph augmentations are indeed unsuitable for circuits. The proposed positive/negative augmentations seem physically grounded (preserving function vs. altering function) and provide a principled way to generate pairs for contrastive learning.

**Weaknesses:**

Limited Diversity of Base Circuits: The pre-training dataset is generated by augmenting only 55 initial circuit topologies. While the dual-augmentation creates vast structural variations around these base topologies, the model might still be biased towards the fundamental structures present in these 55 circuits. It's unclear how well it would generalize to circuits with entirely novel core topologies not seen during pre-training.

Scalability of Graph Construction: While transistor-level representation is general, it can lead to very large graphs for complex digital circuits (which might contain millions or billions of transistors). The paper doesn't discuss the scalability limits of the graph construction or the GNN processing for such large-scale circuits. Gate-level abstractions might still be necessary for computational tractability in those cases.

Rationale for Negative Augmentation Rules: The specific rules for negative augmentation (Figure 3a, Figure 8) seem somewhat ad-hoc. While the goal is to break functionality, the choice of replacement subgraphs could influence what the model learns. A more detailed justification or sensitivity analysis regarding these rules would be helpful.

Comparison to Gate-Level Pre-training: The paper motivates the need for transistor-level analysis but doesn't directly compare DICE against state-of-the-art gate-level pre-trained models (like FGNN2 or others mentioned in Sec 2.2) on purely digital downstream tasks. Such a comparison would help quantify the trade-offs between the finer granularity of DICE and the potentially more abstracted, higher-level features learned by gate-level models.

**Questions:**

Generalization to Novel Topologies: How dependent is DICE's performance on the diversity of the initial 55 circuit topologies used for augmentation? If tested on downstream tasks involving circuits with core structures fundamentally different from these 55, does the performance degrade significantly?

Scalability Concerns: What are the practical limits on the size (number of transistors/nodes) of circuits that DICE can handle during pre-training and inference, given typical GPU memory constraints? How does the computational cost compare to gate-level GNNs for large digital designs?

Impact of Augmentation Rules: Could the authors provide more insight into the design choices for the negative augmentation rules? How sensitive is the pre-training outcome to variations in these rules? Have alternative negative augmentation strategies been considered?

Transistor vs. Gate Level for Digital Tasks: For downstream tasks involving purely digital circuits (like potentially Task 1 similarity or Task 2 delay), how does the performance of fine-tuned DICE compare to models pre-trained directly at the gate level using digital-specific augmentations (e.g., logic synthesis tools)? Does the transistor-level structural information offer advantages even when gate-level abstractions are available?

---

### Official Review · Reviewer_pXN6 · 2025-10-29

**Soundness:** 2
**Presentation:** 2
**Contribution:** 1
**Rating:** 2
**Confidence:** 5

**Summary:**

The paper proposes to build a pre-trained graph-based model for integrated circuits and apply it to address several downstream tasks through fine-tuning.

**Strengths:**

The idea of pre-training a foundation graph-based model for integrated circuits is ambitious and looks exciting.

**Weaknesses:**

The novelty is limited. Unsupervised learning, transistor-level representation, and diverse circuit topologies have been covered by existing methods, such as [1] LaMAGIC2: Advanced Circuit Formulations for Language Model-Based Analog Topology Generation; https://openreview.net/forum?id=Y0zXGw0GUk
[2] AnalogGenie: A Generative Engine for Automatic Discovery of Analog Circuit Topologies
https://openreview.net/forum?id=jCPak79Kev&trk=feed_main-feed-card_feed-article-content


The graph modeling of circuits appears to be inaccurate. Transistors have multiple pins; why use a single node to present them? How is the position information of different devices captured? In addition, if a device-level modeling is used, how can it be fine-tuned to address the digital gate-level task?

Many fine-tuning details are ignored, which makes the results unconvincing. For example, it is nontrivial to fine-tune a purely topology-based model to predict the performance of analog circuits that are determined by device parameters.

The scalability and generalization of the model are not discussed.

**Questions:**

What is the dataset used in the experiment? How do you collect it? What is the circuit distribution and scale of your dataset? What is the size of your model? How do you evaluate the quality of your pre-trained model?

---

### Official Review · Reviewer_xxNM · 2025-11-01

**Soundness:** 2
**Presentation:** 3
**Contribution:** 3
**Rating:** 4
**Confidence:** 4

**Summary:**

This paper proposes DICE (Device-level Integrated Circuits Encoder), a graph neural network pretrained at the transistor level using self-supervised contrastive learning. The key contribution is a dual-augmentation technique that generates diverse circuit topologies through positive augmentation (adding identical devices in series/parallel, preserving functionality) and negative augmentation (replacing subgraphs to alter functionality). The authors claim this increases structural diversity by over 10,000× compared to prior work. DICE is evaluated on three downstream tasks: circuit similarity prediction, delay prediction for digital circuits, and op-amp performance prediction for analog circuits, demonstrating consistent improvements over baseline methods including ParaGraph and DeepGen.

**Strengths:**

+ Good presentation: Clear figures, well-structured paper, and comprehensive appendix with implementation details.
+ Interesting augmentation strategy: The dual-augmentation technique is interesting. Positive augmentation preserves functionality while adding somewhat limited structural diversity; negative augmentation increases functional diversity.

**Weaknesses:**

- Scalability: Adding devices to the graphs through positive or negative augmentation inevitably makes the graph larger. How does it impact the scalability of this method?
- Diversity claim is unclear: "10,000× more topologies" claim is misleading - you're generating 4,000 augmentations per circuit but these aren't functionally unique. The actual functional diversity increase from negative augmentation is unclear, as many negative augmentations may result in non-functional circuits.
- Transistor-level representation motivation is unclear: Why do we need a transistor-level representation for a digital circuit, considering it can be represented at the gate level?
- Generalization concern: Since op-amp topologies from Task 3 appear in the pretraining set (you mention "op-amp circuit topologies used in Task 3 are included in the pretraining dataset"), isn't this somewhat circular? How would performance differ on completely held-out/unseen topologies?
- Dataset transparency: No details about circuit complexity (number of transistors, graph sizes).

**Questions:**

See weakness.

---

### Official Review · Reviewer_HdAg · 2025-11-01

**Soundness:** 2
**Presentation:** 3
**Contribution:** 2
**Rating:** 2
**Confidence:** 3

**Summary:**

The authors propose DICEm which is a dual augmentation strategy that can be applied to any type of circuits. It creates two types of augmentations, positive and negative, and is used in contrastive learning.

**Strengths:**

* The paper is well written, easy to follow. Figures are helpful.
* The authors tried to solve a very important and difficult problem.
* Authors performed various experiments including albation studies.

**Weaknesses:**

* The authors need to explain the motivation and reason for designing model as in Figure 4. There are lots of ways of adopting pretrained models to solving downstream task. But instead of just finetuning the model, the authors propose to use parallel GNNs.
* Hyperparameter search space is quite extensive
* It appears that the experiments were conducted only once and the performance was reported based on that single run. To ensure that the model’s performance is not dependent on a specific random seed but is statistically meaningful, it is necessary to repeat the experiments multiple times and report the mean and standard deviation of the performance.
* I'm not sure whether positive augmentation is valid, as adding circuit component in parallel or in series can still change the operation property of circuit. Authors claim to ignore parameters of circuit components, however, it can still be different as transistors are sensitive to the voltage difference between ports.

**Questions:**

* Identical to 'Weakness' part.

---

### Note · Authors · 2025-11-12

I have read and agree with the venue's withdrawal policy on behalf of myself and my co-authors.